

# Role of NAO and ENSO in the anomalous precipitation in the southern part of China—study on the two contrary high impact weather and climate cases

Qiuxia Wu

State Key Laboratory of Severe Weather, Chinese Academy of Meteorological Sciences, Beijing 100081, China

Correspondence to: Qiuxia Wu (qiuxiawu@camscma.cn)

**Abstract.** Their economic and social importance emphasized by the survey of Department of Disaster Relief, Ministry of Civil Affairs of the People's Republic of China, two different typical patterns of precipitation anomaly in the southern part of China during the 1982/1983 and 2009/2010 cold seasons coincided with the canonical El Niño and positive North Atlantic Oscillation (NAO) and with the El Niño Modoki and negative NAO, respectively. A better understanding of how a particular type of El Niño and a specific phase of NAO worked together to cause the relevant anomalous atmospheric circulation over the East Asia in the two high impact weather and climate cases was an interesting issue and could improve the prediction skill of natural hazards to a certain extent. In conclusion, superimposing on the remote and local Rossby wave responses in the atmosphere induced by the El Niño Modoki-related condensational heat sink over the South China Sea, the downstream extension of the negative NAO was well established by a NAO-induced stationary Rossby wave train along the Asian subtropical jet and played a major role in the persistent dry conditions in the Southwest China for the 2009/2010 boreal winter. On the contrary, for the 1982/1983 boreal winter, the canonical El Niño weakened the downstream extension of the positive NAO, and induced by the canonical El Niño-related condensational heat sink over the western equatorial Pacific Ocean, the remote and local Rossby wave responses in the atmosphere played a leading role in the sustained wet conditions in the South China.

## 1. Introduction

A severe drought, attributed to a prolonged shortage of precipitation, occurred in the Southwest China during the 2009/2010 cold season (Lu et al., 2011; Huang et al., 2012; Barriopedro et al., 2012). On the contrary (see Fig. 1), an abundant precipitation occurred in the South China during the 1982/1983 cold season (Lau and Sheu, 1988; Wang et al., 2000; Chen, 2002). Being orthogonal to each other, the first and the fourth principal components of precipitation field (accounting for 18% and 4% of the total variance, statistically significant in terms of the sampling error bars according to the rule proposed by




North et al. (1982)) respectively were similar to the anomalous precipitation patterns of the 1982/1983 and the 2009/2010 cases (see Fig. 2). Therefore, the two cases could represent two different typical patterns of precipitation anomaly in China; their economic and societal importance is emphasized by the survey of Department of Disaster Relief, Ministry of Civil Affairs of the People's Republic of China (2010). Coincidently, as the highly anomalous forcing to the atmospheric circulation over the East Asia,

both the El Niño-Southern Oscillation (ENSO, Bjerknes, 1969; Rasmusson and Carpenter, 1982; Philander, 1983) and the North Atlantic Oscillation (NAO, Defant, 1924; Walker and Bliss, 1932; Bjerknes, 1964; Hurrell et al., 2001) were under way.

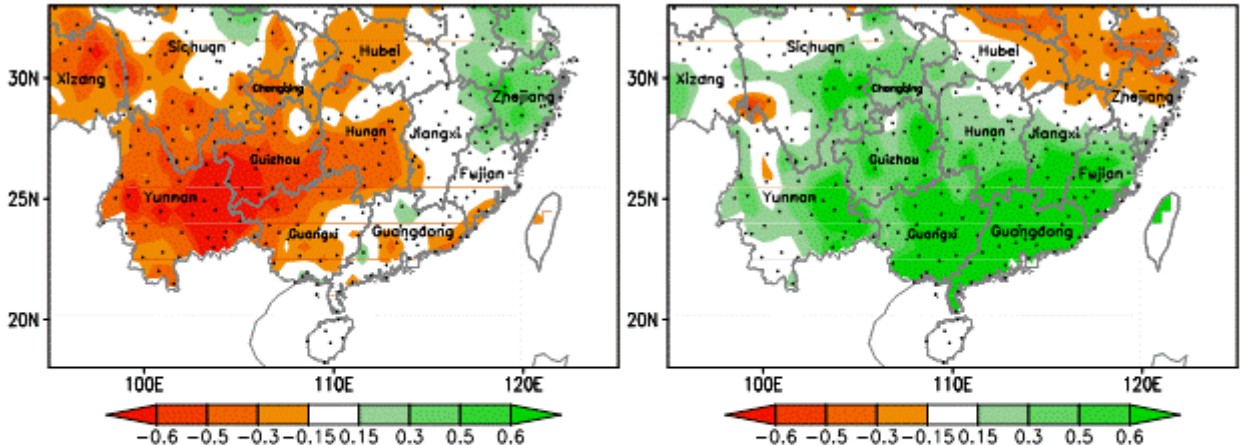

**Fig. 1** Percentage anomaly of the accumulated precipitation from Sep. 2009 to Mar. 2010 (left panel) and from Sep. 1982 to Mar. 1983 (right panel). The black dots denote the spatial distribution of precipitation gauges. The analysis is done using the historical archives of the observed 752-station precipitation in China (Jan. 1951 to present)

As the dominant global climate signal (Trenberth, 1976; Weare et al., 1976; Julian and Chervin, 1978;
Horel and Wallace, 1981; Arkin, 1982; Pan and Oort, 1983; Rasmusson and Wallace, 1983), the El Niño-related sea surface temperature anomaly (SSTA) is associated with suppressed convection over the western equatorial Pacific Ocean, and the induced condensational heat sink works as an external force to excite local and remote Rossby wave responses in the atmospheric circulation over the East Asia (Matsuno, 1966; Webster, 1972; Gill, 1980; Webster, 1981; Trenberth et al., 1998). In the vicinity of the
heat sink, an anomalous Philippine Sea anticyclone develops in the lower troposphere, and in the midlatitude westerlies, an upper troposphere anomalous cyclone develops over the East Asia. Attributed to the above well-defined circulation pattern, the precipitation anomaly has one positive center in the





South China to the southern part of Japan during the fall of an El Niño developing year through the
following spring (Tao and Zhang, 1998; Lau and Nath, 2000; Wang et al., 2000; Zhang and Sumi, 2002;
Wu et al., 2003). This feature is distinct in the 1982/1983 case in the South China (He et al., 2006),
coinciding with the strongest canonical El Niño (the eastern-Pacific type of El Niño, Trenberth and
Stepaniak, 2001).

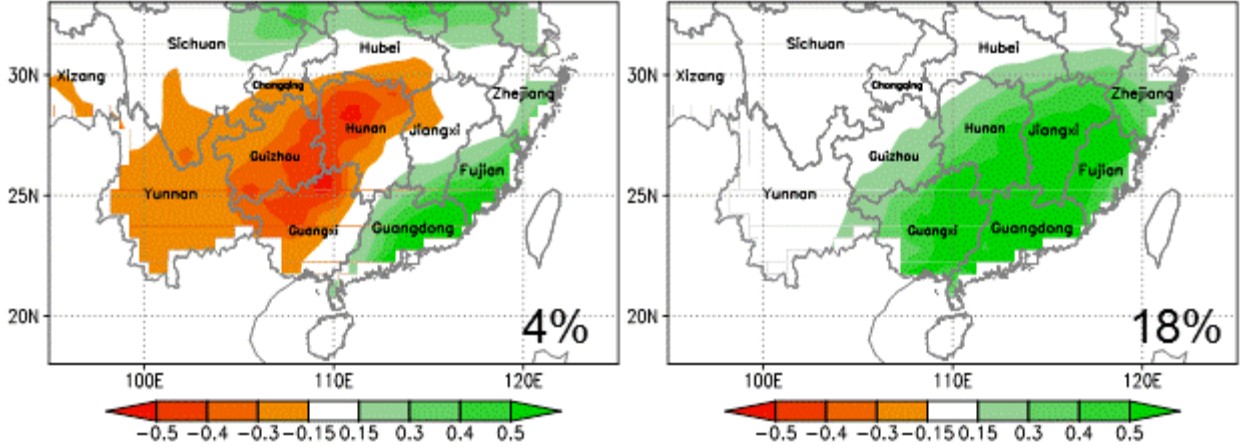

**Fig. 2** The fourth (left panel) and first (right panel) principal components of precipitation
anomaly. The analysis is done using the monthly grid precipitation based on the Chen's daily
grid precipitation in China at 0.5° × 0.5° resolution grid (Jan. 1961 to Dec. 2010)

However, a central-Pacific type of El Niño (so-called the El Niño Modoki, Ashok et al., 2007; Ashok
and Yamagata, 2009; Kao and Yu, 2009; Kug et al., 2009; Lee and McPhaden, 2010) has a different
effect on the East Asian winter climate (Weng et al., 2009; Feng et al., 2010; Kim et al., 2012; Yuan and
He, 2013). During the El Niño Modoki, the condensational heat sink is nearer to the South China Sea
(SCS), the induced lower troposphere anomalous anticyclone develops over the SCS to the eastern
Indian Ocean, as pointed out by Weng et al. (2009); the upper troposphere anomalous cyclone over the
East Asia develops nearer to the Middle East, accompanied with a weak anomalous ridge of high
pressure predominating over the Indochina Peninsula. Consequently, the westward movement of the
induced anomalous atmospheric circulation system tends to cause a different precipitation anomaly
pattern in the East Asia. Coinciding with a strong El Niño Modoki, the 2009/2010 severe drought in the
Southwest China showed such a different pattern.
The NAO is the prominent and recurrent pattern of variations in the atmospheric circulation over the
Northern Hemisphere (Wanner et al., 2001; Marshall et al., 2001; Hurrell et al., 2003). Its eastward



extension as far as the Mediterranean Sea leads to an optimal vorticity source for effectively setting up a quasi-stationary Rossby wave train along the Asian subtropical jet (Watanabe 2004). In this respect, similar in pattern in the upper troposphere to the Arctic Oscillation, the negative NAO has a downward extension and is associated with an anomalous cyclone predominating over the Middle East, ridge of high pressure over the Indochina Peninsula, and cyclone over the North and Northeast China; at the same time, the anomalous cyclone over the North and Northeast China also develops in the lower troposphere. Under the circumstances, the Southwest China tends to have a precipitation shortage due to a divergence of water vapor and a strengthened northerly wind carrying drier air. Actually, the significant relationship between the negative NAO and the precipitation shortage in the Southwest China in boreal winter is emphasized by Xu et al. (2012).

Note that the negative NAO has a preferred occurrence during the El Niño and the positive NAO during the La Niña (Pozo-Vázquez D. et al., 2001; Li and Lau, 2012). Accordingly, the occurrence and downstream extension of NAO could be modulated by the ENSO to a certain extent. Thus, we assumed that the NAO worked together with the ENSO to make an anomalous weather and climate over the East Asia when they are concurrent.

For the 2009/2010 drought period, the Southwest China was influenced by an anomalous ridge of high pressure in the upper troposphere and an induced strong subsidence of air according with a convergence of air in the upper troposphere (see the left panels of Fig. 3). At the same time, an anomalous cyclone developed over the North and Northeast China, and a lower troposphere anomalous anticyclone circulation developed over the Maritime Continent to the eastern Indian Ocean (see the upper panel of Fig. 4). As a result, the dry conditions in the Southwest China occurred with a weakened water vapor flux from the Bay of Bengal (BOB), a divergence of water vapor and a strengthened northerly wind carrying drier air. As pointed out in the previous research (Yang et al., 2012), the relevant anomalous atmospheric conditions throughout the East Asia are attributed to both the negative NAO and the El Niño Modoki. However, no anyone did a thorough study on how the two factors worked together to set up the relevant large-scale atmospheric circulation anomaly, which was focused on in the study.

The 1982/1983 wet case in the South China coincided with both the canonical El Niño and the positive NAO (Rogers, 1984). The relevant anomalous atmospheric circulation was an upper troposphere anomalous cyclone predominating over the East Asia and a zonal stretched anomalous Philippine Sea anticyclone developing in the lower troposphere (see the right panels of Fig. 3 and the lower panel of Fig. 4), and both of them had an influence on the anomalous atmospheric conditions in the South China. As a result, a strong rise of air according with a upper troposphere anomalous divergence of air worked together with a strengthened water vapor flux from the BOB and SCS to lead to a convergence of water vapor and cause the wet conditions in the South China (see the lower panel in Fig. 4). Motivated by the


contrary anomalous moist conditions of the two cases, the analysis was made to contrast the combined impact of the El Niño Modoki and the negative NAO with the one of the canonical El Niño and the positive NAO on the East Asian weather and climate.

The data and methodology was described in section 2. In section 3, the impact of the ENSO on the East
Asian atmospheric circulation as well as its role in the 2009/2010 and 1982/1983 cases were shown. The downstream extension of the NAO impact and its physical connection to the relevant atmospheric circulation anomalies throughout the East Asia for the two cases were stated in section 4, and the potential modulation of the NAO impact by the ENSO was also discussed in this section. Finally, section 5 provided a summary and briefly discussed other potential factors.

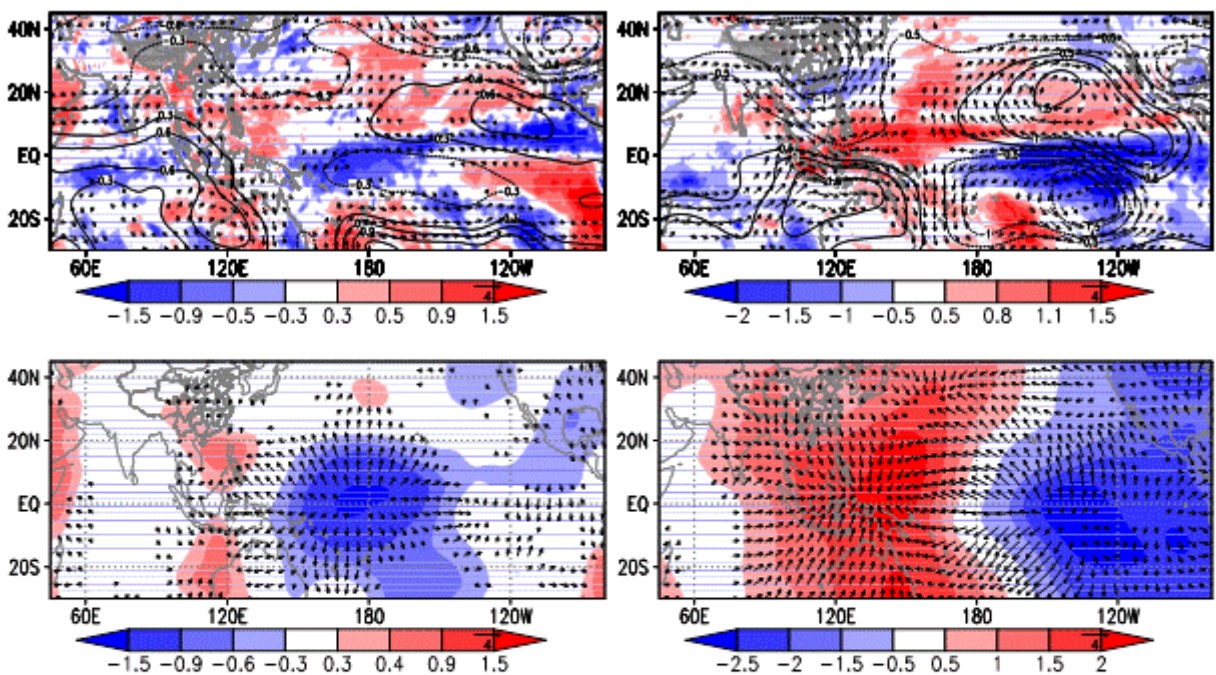

**Fig. 3** The composite 600hPa vertical velocity anomaly (ω, color shading), 200hPa streamfunction and rotational wind anomaly (contour and vector) for Sep. 2009-Mar. 2010 (upper left panel) and for Sep. 1982-Mar. 1983 (upper right panel). Lower panels show the composite 200hPa velocity potential and divergent wind anomaly (color shading and vector) for Sep. 2009-Mar. 2010 (lower left panel) and for Sep. 1982-Mar. 1983 (lower right panel). The Tibetan Plateau is shown in grey shading. The anomaly is standardized. The analysis is
done using the Era-Interim reanalysis (Jan. 1979 to present)


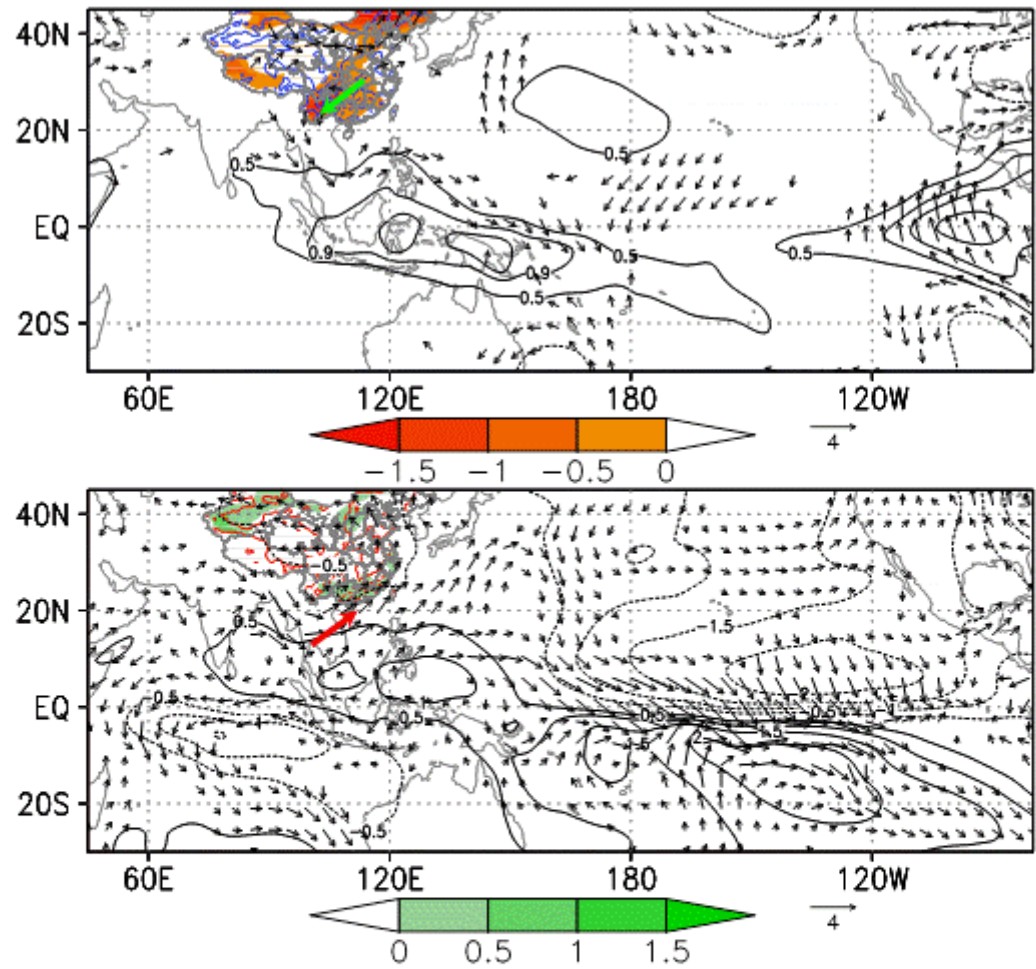

**Fig. 4** The composite precipitable water anomaly (color shading), mass-weighted vertical integral of water vapor flux anomaly and its convergence (vector and color contour), and 850hPa streamfunction anomaly (black contour) for Sep. 2009-Mar. 2010 (upper panel) and for Sep 1982-Mar. 1983 (lower panel). The thick arrow denotes the direction of anomalous water vapor flux to the southern part of China. The anomaly is standardized. The analysis is done using the Era-Interim reanalysis (Jan. 1979 to present)

## 2.  Data and methodology



## 2.1 Data

The data used here were the monthly NOAA Extended Reconstructed Sea Surface Temperature V3b (ERSST.v3), the monthly ERA-Interim and National Centers for Environmental Prediction/National Center for Atmospheric Research (NCEP/NCAR) Reanalysis, which could give a reasonable and comprehensive depiction of the atmospheric and underlying surface conditions (Kalnay et al., 1996; Trenberth et al., 2000; Smith et al., 2008; Dee et al., 2011). In addition, the Chen's daily grid

precipitation in China was used to calculate the monthly grid precipitation (Chen et al., 2010), and as shown in Fig. 2, the monthly grid precipitation was used to make the principal components analysis to highlight the statistically significant modes of precipitation anomaly in China. For each physical variable, the climatology was defined as the 30-year average (1980-2009) of each calendar month, and the anomalous fields were calculated by subtracting the climatology from the raw time series. Using the

Chinese 752-station historical archives of the observed 24-h accumulated precipitation (January 1951 to the present), an accumulated anomalous precipitation for a particular span of time could be exactly demonstrated by the percentage anomaly of precipitation defined as:

$$P_{r_{sum}}' = \frac{P_{r_{sum}} - \overline{P_{r_{sum}}}}{\overline{P_{r_{sum}}}} \tag{1}$$

where $P_{r_{sum}}$ represented the accumulated precipitation for a particular span of time, and $\overline{P_{r_{sum}}}$

denoted the 30-year average (1980-2009) of the accumulation. Parallel analyses with the ERA-Interim and the NCEP/NCAR Reanalysis were carried out to confirm the final conclusion, and only the analysis with the ERA-Interim Reanalysis was shown.

## 2.2 Methodology

Closely associated with the persistent anomalous atmospheric circulation, the wave activity flux $\vec{C}$ for

stationary Rossby waves on a real basic flow was calculated using the formulation of Eq. (C5) in Takaya et al. (2001) as follows:

$$\begin{pmatrix} C_x \\ C_y \end{pmatrix} = \frac{1}{2|\vec{U}|} \begin{pmatrix} U(\psi_x'^2 - \psi'\psi_{xx}') + V(\psi_x'\psi_y' - \psi'\psi_{xy}') \\ U(\psi_x'\psi_y' - \psi'\psi_{xy}') + V(\psi_y'^2 - \psi'\psi_{yy}') \end{pmatrix} \tag{2}$$



where a steady zonal inhomogeneous basic flow $\vec{U} = (U,V)^T = (-\Psi_y, \Psi_x)^T$(the superscript $^T$ indicated

the vector transposition) was the geostrophic velocity. The streamfunction $\psi = \Psi(x,y,z) + \psi'$, and $\psi'$

represented the anomalous field. The NAO-related wave activity flux was calculated with the regression streamfunction anomaly on the monthly CPC standardized NAO index. The regression streamfunction anomaly on the NAO index could be taken as the typical spatial pattern of anomalous atmospheric circulation related to the NAO. The potential impact of the NAO was also demonstrated by the Pearson correlation of the relevant physical fields with the NAO index and the semi-partial Pearson correlation by removing the impact of the ENSO.

Using the ERSST anomaly, the NINO3 index and the El Niño Modoki index (EMI) were calculated to represent the temporal evolutions of the canonical El Niño and the El Niño Modoki (Trenberth, 1997; Ashok et al., 2007). We calculated the Pearson correlation of the velocity potential anomaly with each index. Accordingly, the PEMI ($105^0$E-$135^0$E,$10^0$N-$30^0$N) and the PNINO3 ($107.5^0$E-$160^0$E,$10^0$S-$20^0$N) index were defined as the area-averaged 200hPa velocity potential anomaly in the two regions. The two original indices could represent the temporal evolutions of the condensational heat sink-induced divergence circulation over the SCS during the El Niño Modoki and over the western equatorial Pacific Ocean during the canonical El Niño. Then, the Pearson correlation of the relevant physical fields with the two original indices was calculated to reveal the impact of the canonical El Niño and the El Niño Modoki. Also, their impact on the atmospheric circulation was demonstrated with the regression streamfunction anomalies on the two original indices. The regression fields could be taken as the typical spatial patterns of anomalous atmospheric circulation related to the two types of ENSO.

To determine how well the 2009/2010 drought pattern matched the typical patterns related to the El Niño Modoki and the negative NAO, and the 1982/1983 wet pattern matched the typical patterns related to the canonical El Niño and the positive NAO, the Taylor diagram was used to demonstrate the correspondence between two patterns (Taylor, 2001). In this diagram, the four statistics, i.e. the correlation coefficient, Root-Mean-Square (RMS) difference between two fields, and standard deviations of each field, provide the complementary statistical information quantifying the similarity between the two patterns. Additionally, interpreted identically as a Pearson correlation (Cook et al., 2010), the correlation coefficient has the relationship with the other three statistics named the Law of Cosine. The pattern correlation (i.e. the Taylor diagram) was calculated over the region $50^0$E-$170^0$E and EQ-$37.5^0$N. This area encompassed the affected region of both the ENSO and the NAO, extending from the Middle East to the East Asia.



## 3. ENSO impact

For the 2009/2010 drought case, the prolonged shortage of precipitation lasted from Mid-August 2009 to March 2010. Responsible for the 1982/1983 wet case, the large-scale atmospheric circulation in September 1982 made a transition to favor the wet conditions which lasted until April 1983. As the highly anomalous forcing to the atmospheric circulation over the East Asia, both the NAO and the ENSO had the significant magnitude for the period of November to the following February in the two cases. In order to stress the important role of the ENSO and the NAO in the relevant atmospheric circulation anomalies over the East Asia, the following analysis focused on the boreal winter months (December to February, DJF).

### 3.1 Canonical El Niño

As shown in the lower right panel of Fig. 5, the canonical El Niño-related SSTA was associated with the convergence of air in the upper troposphere over the western equatorial Pacific Ocean, essentially identical to the pattern for the period of the 1982/1983 case (see the lower right panel of Fig. 3).

To demonstrate the impact of the canonical El Niño, the upper right panel of Fig.5 shows the Pearson correlation of streamfunction anomaly with the PNINO3 index. To the northwest flank of the canonical El Niño-related condensational heat sink, an upper troposphere anomalous cyclone predominated over the East Asia, indicating a remote Rossby wave response in the atmosphere. Adjacent to the south of the anomalous cyclone, a lower troposphere anomalous anticyclone with a zonal stretched shape-the anomalous Philippine Sea anticyclone was located over the Philippine Sea to the SCS, corresponding to a local Rossby wave response. The anomalous Philippine Sea anticyclone strengthened a water vapor flux from the BOB and SCS to the southern part of China where, at the bottom of the upper troposphere anomalous cyclone (i.e. at the upper troposphere trough), the induced convergence of water vapor tended to cause the wet conditions (see the middle right panel of Fig.5).

To assess the canonical El Niño's impact on the 1982/1983 wet case, the pattern correlation was analyzed between the composite regression streamfunction anomaly on the PNINO3 index and the composite observation for the 1982/1983 boreal winter (see Fig.6).

On the polar style graph, the REF points represented the observed fields, and the PNINO3 points denoted the regression fields on the PNINO3 index. The radial distance from the origin to the point was proportional to the standard deviation of each field (referring to the amplitude of variation in each field). Denoting the phase and structure difference between the two fields, the azimuthal position of the PNINO3 point represented the correlation coefficient with the observed field, and the distance between the two points indicated the RMS difference. The pattern correlation demonstrated that the relevant atmospheric circulation anomaly for the 1982/1983 case bore a strong resemblance to the typical spatial



Natural Hazards
and Earth System
pattern related to the canonical El Niño, both in the phase and structure and in the amplitude of variation. With amplitude of variation identical to the 1982/1983 pattern, however, the canonical El Niño-related anomalous anticyclone in the lower troposphere was located over the Philippine Sea to the SCS and

shifted westward about $20^0$ longitudes (see the middle right panel of Fig. 6). The westward shift in the phase and structure also occurred for the upper troposphere pattern (see the upper right panel of Fig. 6). In addition to a smaller RMS difference from the observed field, the anomalous atmospheric circulation in the upper troposphere was subject to a more significant influence of the canonical El Niño than the one in the lower troposphere. According to the strong similarity between the two fields, the strong

canonical El Niño might play an important role in the relevant atmospheric circulation anomaly over the East Asia for the 1982/1983 wet case.

## 3.2    El Niño Modoki

Differing from the canonical El Niño, the El Niño Modoki-related convergence of air in the upper troposphere moved northwestward and was located over the SCS (see the lower left panel of Fig. 5),

identical to the spatial pattern for the 2009/2010 drought period (see the lower left panel of Fig. 3). Likewise, we calculated the Pearson correlation of streamfunction anomaly with the PEMI index (see the upper left panel of Fig. 5). Associated with the northwestward movement of the condensational heat sink, the aforementioned upper troposphere anomalous cyclone actually moved westward and was located nearer to the Middle East. At the same time, the anomalous Philippine Sea anticyclone also

moved westward and was located over the SCS to the East Indian Ocean. In other words, the induced remote and local Rossby wave responses in the atmosphere revealed a westward phase shift from the ones related to the canonical El Niño (see the upper right panel of Fig. 5). It was interesting to note that a weak anomalous ridge of high pressure in the upper troposphere appeared over the Indochina Peninsula. Under the circumstances, compared with the states during the canonical El Niño, the

convergence of water vapor in the southern part of China and the water vapor flux from the BOB and the SCS was greatly weakened (see the middle left panel of Fig. 5).

To assess the role of the El Niño Modoki in the 2009/2010 drought, the pattern correlation was analyzed for the 2009/2010 boreal winter (see Fig. 7). On the polar style graph, the PEMI points represented the regression fields on the PEMI index. The radial dashed line from the origin through the PEMI point was

labeled by the correlation coefficient with the observed field, equal to the cosine of the angle made with the abscissa, denoting a phase shift from the observation.

In the lower troposphere, the El Niño Modoki-related anomalous anticyclone over the SCS was nearly in phase with the one for the 2009/2010 boreal winter and had a more zonal stretched shape; however, the observed anomalous cyclone over the North and Northeast China might not have a link to the impact

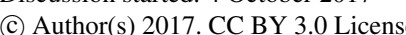


of the El Niño Modoki (see the middle right panel of Fig. 7). In the upper troposphere, the observed

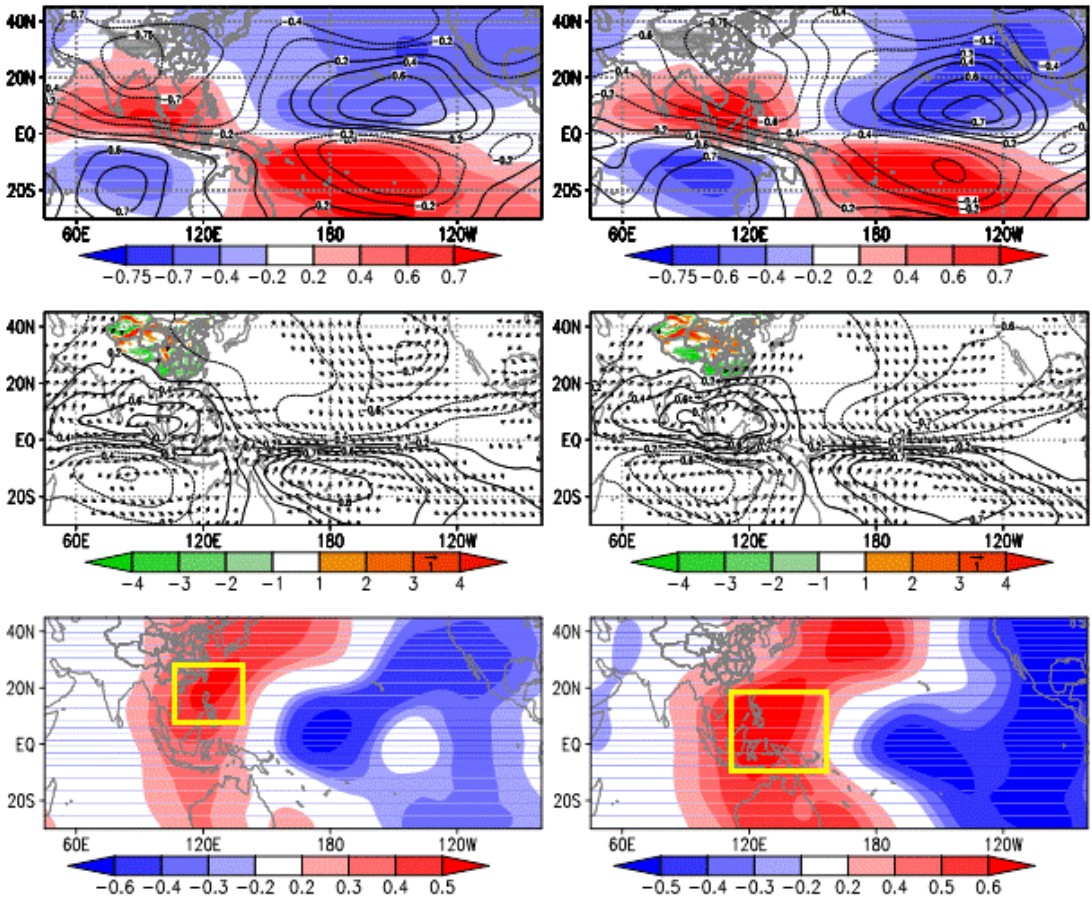

**Fig. 5** Pearson correlation in December-January-February (DJF) of streamfunction and column-integrated water vapor flux anomaly with the PEMI index (upper and middle left panels) and with the PNINO3 index (upper and middle right panels). In the upper panels, the color shaded contour denotes the 850hPa streamfunction, and the black contour represents the 200hPa streamfunction. In the middle panels, the vectors denote the column-integrated water vapor flux and its convergence is normalized and represented by the color shaded contour, and the Pearson correlation of the 850hPa streamfunction anomaly is also added (black contour). The lower panels show the Pearson correlation in DJF of the 200hPa velocity potential anomaly with the EMI index (lower left panel) and with the NINO3 index (lower right panel). Yellow rectangle denotes the region of the PEMI index (lower left panel) and the PNINO3 index (lower right panel). Only the region at 95% significant confidence level (p < 0.05) is contoured. The analysis is done using the ERSST v.3 (Jan. 1854 to present) and the Era-Interim reanalysis (Jan. 1979 to present)


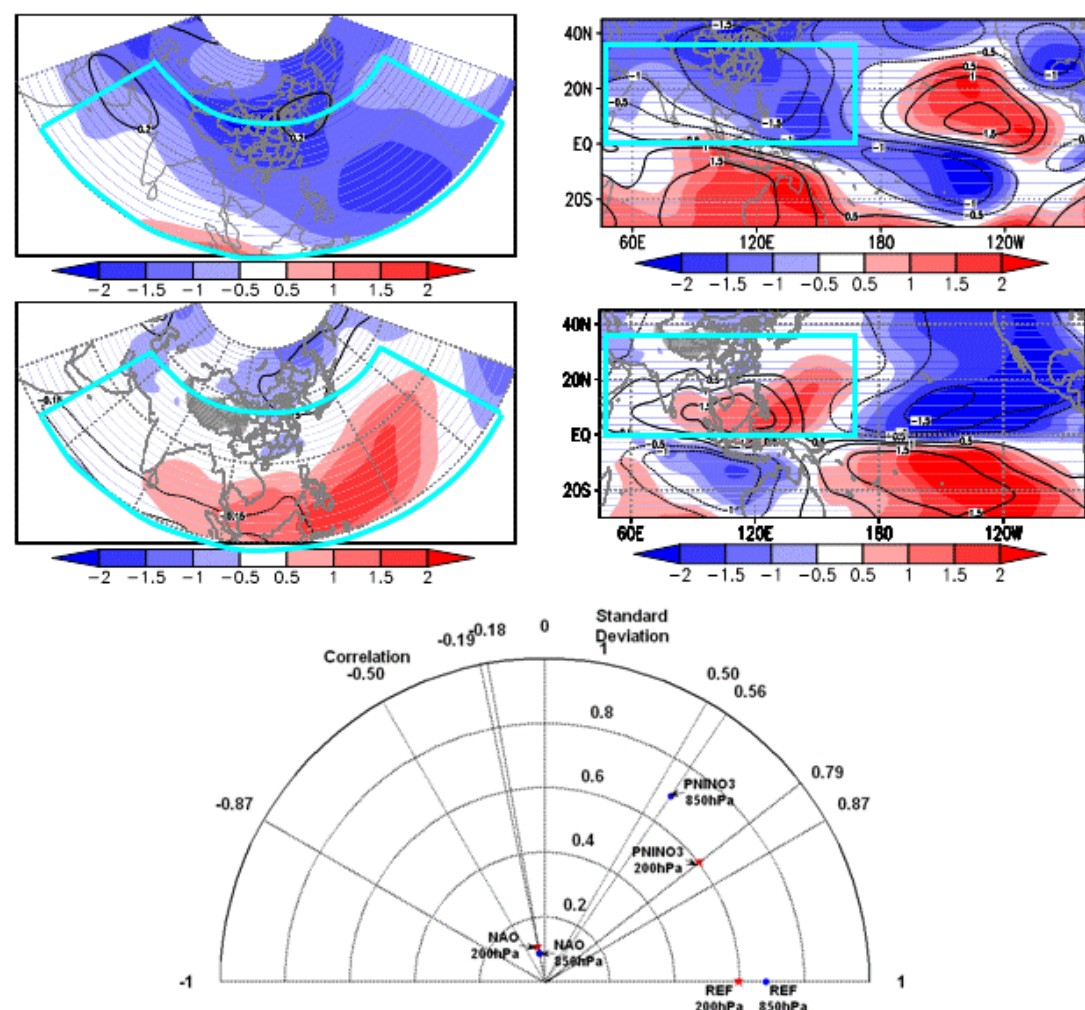

**Fig. 6** The Taylor diagram (lower panel) of the composite regression 200hPa and 850hPa streamfunction anomalies on the NAO and PNINO3 index and the composite observed fields for Dec. 1982-Feb. 1983. The NAO and PNINO3 points represent the regression fields on the two indices. The observed fields are denoted by the REF points. Upper panels show the composite observed 200hPa streamfunction anomaly (color shaded contour) and the composite 200hPa streamfunction anomaly (black contour) on the NAO (upper left panel) and PNINO3 index (upper right panel) for Dec. 1982-Feb. 1983. Middle panels are same to the upper panels except for the 850hPa fields. The light blue fan-shaped and rectangular area denotes the region for the pattern correlation analysis. All the streamfunction anomalies used here are standardized with the variance of the observed streamfunction anomaly in DJF during 1979 to the present. The analysis is done using the CPC standardized NAO (Jan. 1950 to present) and Era-Interim reanalysis (Jan. 1979 to present)



anomalous cyclone predominating over the Middle East had a westward phase shift of about $20^0$ longitudes from the El Niño Modoki-related one, but the observed anomalous ridge of high pressure

over the Indochina Peninsula was in phase with and stronger than the one related to the El Niño Modoki (see the upper right panel of Fig. 7). Even though a nearly identical RMS difference from the REF points both in the upper and lower troposphere, a smaller phase shift from the observed 200hPa field demonstrated a better similarity and confirmed a more significant influence of the El Niño Modoki on the upper troposphere atmospheric circulation over the East Asia (see the lower panel of Fig. 7). In

conclusion, the much smaller than $90^0$ phase angle from the observed fields suggested that the El Niño Modoki made a substantial contribution to but did not fully determine the phase and structure of the anomalous atmospheric circulation responsible for the 2009/2010 drought. In addition, the standard deviations of the PEMI points were less than two-thirds the ones of the REF points, indicating that the amplitude of variation in the atmospheric circulation anomaly related to the El Niño Modoki was less

than two-thirds the observed one. Note that the El Niño Modoki-related upper troposphere anomalous ridge of high pressure over the Indochina Peninsula was too weak to be comparable with the observation. Inferred from these features, a strong negative NAO might play a major role in strengthening and maintaining the relevant atmospheric circulation anomaly over the East Asia for the 2009/2010 drought.

**4.   NAO impact**

As shown in the lower left panel of Fig. 8, for the 2009/2010 drought period, the southern end of the negative NAO steadily predominated over the North Atlantic Ocean stretching eastward as far as the Mediterranean Sea, and the corresponding divergence circulation stretched eastward from the North Atlantic Ocean. Meanwhile, the entrance of the Asian subtropical jet moved from its normal position to

the north nearer to the Mediterranean Sea, and a continuous eastward wave activity flux was confined on the Asian subtropical jet (see the upper left panel of Fig. 8). According to Watanabe (2004), under the circumstances, a physical connection was established between the NAO and the East Asian atmospheric circulation anomaly.

On the contrary, the southern end of the 1982/1983 NAO predominated over the western North Atlantic

Ocean, coinciding with a certain southward shift of the entrance of the Asian subtropical jet (see the right panels of Fig. 8). While with an eastward wave activity flux over the Mediterranean Sea, a continuous eastward wave activity flux to the East Asia did not occur in the composite illustration (see the upper right panel of Fig. 8). This fact suggested that no steadily eastward Rossby wave energy transport from the North Atlantic Ocean and its vicinity was set up to make a substantial contribution to

the relevant anomalous atmospheric circulation over the East Asia for the period of the 1982/1983 wet



case.

**Fig. 7** The same as in Fig. 6 except for the composite regression 200hPa and 850hPa streamfunction anomalies on the NAO and PEMI index and the composite observed fields for Dec. 2009-Feb. 2010

## 4.1 NAO-related Rossby wave train

Generally speaking, the southern end of the NAO predominated over the North Atlantic Ocean and
extended eastward as far as the Mediterranean Sea, coinciding with a divergence circulation stretching

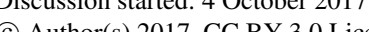


**Fig. 8** The composite 200hPa geopotential height anomaly (color shading, m), zonal wind (green thick solid contour, 30 m s⁻¹) and eastward wave activity flux of stationary Rossby waves (vector, m⁻² s⁻²) (upper left panel), and the composite 850hPa geopotential height anomaly (color shading, m) and divergent wind anomaly (vector, m s⁻¹) (lower left panel) for Sep. 2009-Mar. 2010. The right panels are same to the left panels except for Sep. 1982-Mar. 1983. The analysis is done using the Era-Interim reanalysis (Jan. 1979 to present)





from the North Atlantic Ocean to the Mediterranean Sea (see the upper right panel of Fig. 9). Under the circumstances, an eastward wave activity flux in the upper troposphere over the Mediterranean Sea steadily directed downstream to the East Asia along the Asian subtropical jet, establishing a physical
connection of the NAO with the anomalous atmospheric circulation over the East Asia, and a zonal-oriented stationary Rossby wave train occurred along the Asian subtropical jet (see the upper left panel of Fig. 9).

In the upper troposphere, the negative NAO-related stationary Rossby wave train included an anomalous cyclone predominating over the Middle East, ridge of high pressure over the Indochina
Peninsula, and cyclone over the North and Northeast China. Additionally, the anomalous cyclone over the North and Northeast China also well developed in the lower troposphere (see the upper right panel of Fig. 9). Note that the Southwest China was located on the crest of the anomalous ridge of high pressure over the Indochina Peninsula. This spatial pattern was typical under the impact of the negative NAO and essentially identical to the 2009/2010 drought pattern (see the upper left panel of Fig. 3).
Under the circumstances, the dry conditions in the Southwest China tended to occur coinciding with a divergence of water vapor and an anomalous northerly wind carrying drier air (see the lower panel of Fig. 9).

Similarly, to assess the impact of the NAO on the relevant atmospheric circulation for the two cases, we calculated the pattern correlation over the Middle East to the East Asia (see Fig. 6 and 7). In the lower
panel of Fig. 7, the NAO points represented the composite regression fields on the NAO index for the 2009/2010 boreal winter. Indicated by the more statistically significant correlation coefficient of the NAO points, the 2009/2010 drought pattern had a closer similarity in the phase and structure to the NAO-related typical spatial pattern than the El Niño Modoki-related one. In the upper troposphere, with a standard deviation in a ratio of 2/3 to the observation, the NAO-related Rossby wave train—an
anomalous cyclone over the Middle East, ridge of high pressure over the Indochina Peninsula and cyclone over the North and Northeast China, was nearly in phase with the observation (see the upper left panel of Fig. 7). In the lower troposphere, with a standard deviation nearly half the observation, the NAO-related anomalous cyclone over the North and Northeast China was located slightly to the east about $10^0$ longitudes of the observed one (see the middle left panel of Fig. 7). In addition to a shorter
distance of the NAO points from the REF points than from the PEMI points, the closer resemblance of the NAO-related pattern to the observed pattern confirmed the major role of the negative NAO in the relevant atmospheric circulation anomaly over the East Asia for the 2009/2010 drought case.

To the contrary, corresponding to a nearly $100^0$ phase angle from the 1982/1983 wet pattern, the azimuthal positions of the NAO points denoted that the NAO-related pattern was statistically orthogonal
to the observed fields in the phase and structure (see the lower panel of Fig. 6). Moreover, the NAO

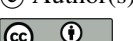


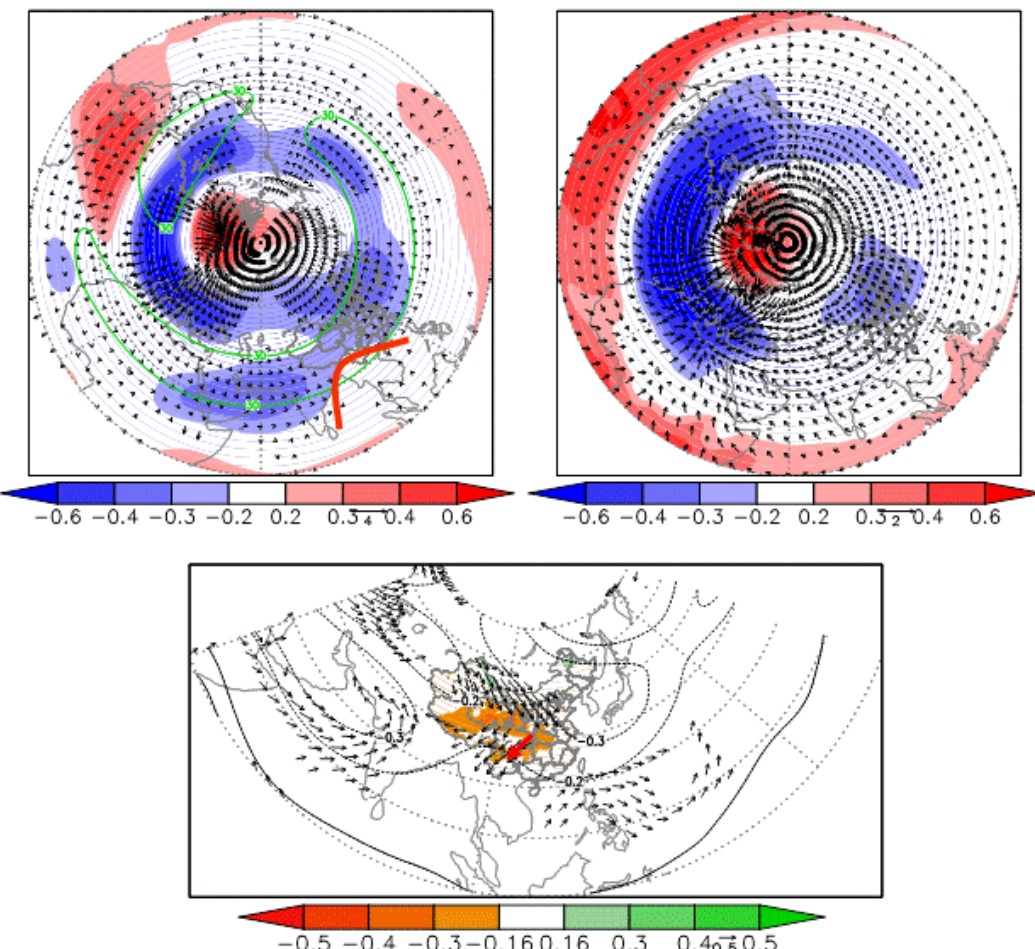

**Fig. 9** The eastward NAO-related wave activity flux (vector) and the Pearson correlation in DJF of the NAO index with the 200hPa streamfunction anomaly (color shading) (upper left panel), the Pearson correlation in DJF of the NAO index with the 850hPa streamfunction and divergent wind anomaly (color shading and vector) (upper right panel), and the Pearson correlation in DJF of the NAO index with the column-integrated water vapor flux and Chen's grid precipitation anomaly (vector and color shading) (lower panel). The Pearson correlation is multiplied by minus one and contoured at 95% significant confidence level ($p < 0.05$). In the upper left panel, the green solid thick contour represents the 30-year average (1980-2009) of the 200hPa zonal wind of 30 m s$^{-1}$ in DJF, and the thick orange curve denotes the anomalous ridge of high pressure with anticyclonic circulation. In the lower panel, the thick arrow represents the direction of anomalous water vapor flux to the southern part of China and the Pearson correlation in DJF of the NAO index with the 200hPa streamfunction anomaly is also added (black contour). The analysis is done using the CPC standardized NAO (Jan. 1950 to present), Chen's daily grid precipitation in China (Jan. 1961 to Dec. 2010) and


points with a large RMS difference from the REF points had standard deviations much smaller than the observations. Accordingly, the positive NAO could not have a significant influence on the atmospheric circulation over the East Asia for the 1982/1983 boreal winter. Also, the finding confirmed the absence

of a continuous eastward wave activity flux to the East Asia for the 1982/1983 case period (see the upper right panel of Fig. 8).

## 4.2    Potential modulation by the ENSO

Inferred from the previous studies and the present research, the negative NAO tended to make a substantial impact on the East Asian atmospheric circulation coinciding with the El Niño Modoki. The

inference was corroborated by the semi-partial Pearson correlation with the NAO index by removing the impact of the ENSO. As shown in Fig. 10, the negative NAO-induced Rossby wave train was not manifest by removing the impact of the El Niño Modoki. Note that, over the Middle East and East Asia, the typical spatial pattern related to the NAO showed a notable resemblance in the phase and structure to the one related to the El Niño Modoki (see Fig. 7). In this regard, coinciding with the El Niño

Modoki, the strengthened downstream extension of the negative NAO might be partly seen as the superimposition of the atmospheric circulation anomaly related to the El Niño Modoki.

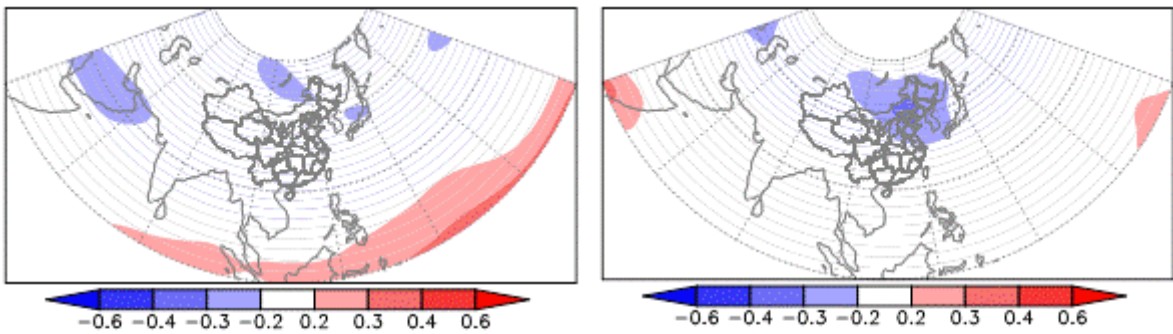

**Fig. 10** Semi-partial Pearson correlation in DJF of the reversed NAO index with the 200hPa (left panel) and 850hPa streamfunction anomaly (right panel) by removing the impact of the El Niño Modoki, contoured at 95% significant confidence level ($p < 0.05$). The analysis is done using the CPC standardized NAO (Jan. 1950 to present) and the Era-Interim reanalysis (Jan. 1979 to present)

Specifically, the southern end of the negative NAO during the 2009/2010 drought period predominated over the North Atlantic Ocean as far as the Mediterranean Sea and exerted a dominant effect to shift the

entrance of the Asian subtropical jet northward nearer to the Mediterranean Sea (see the left panels of Fig. 11). Under the circumstances, the NAO-induced Rossby wave train occurred along the Asian




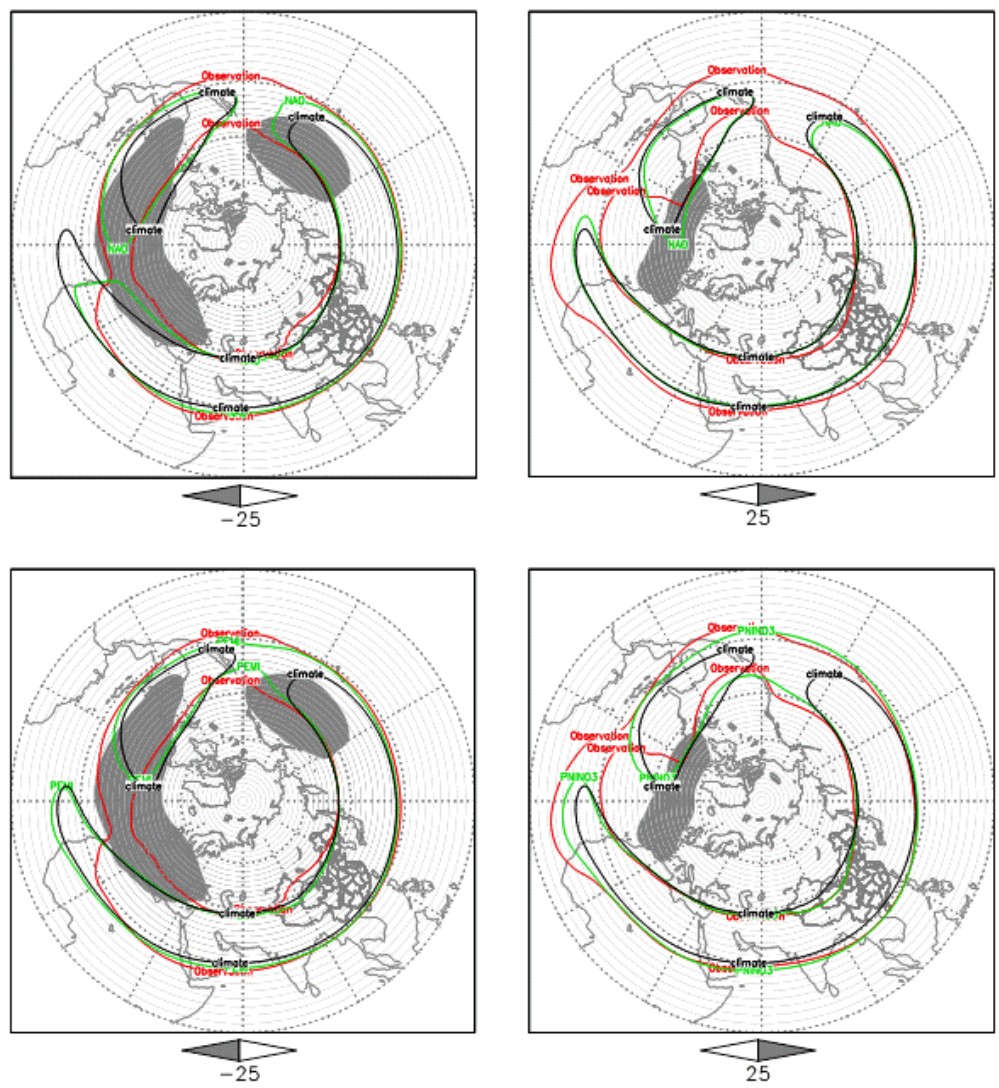

**Fig. 11** The composite regression 200hPa zonal wind on the NAO (upper left panel) and PEMI index (lower left panel) for Dec. 2009-Feb. 2010. For Dec. 1982-Feb. 1983, the composite regression 200hPa zonal wind on the NAO index is shown in the upper right panel and the one on the PNINO3 index in the lower right panel. The red contour denotes the observed zonal wind of 30 m s⁻¹, the black contour represents the 30-year average (1980-2009) of zonal wind of 30 m s⁻¹, and the green contour demonstrates the regression zonal wind of 30 m s⁻¹. The grey shading represents the composite observed geopotential height anomaly (m) at 850hPa pressure level for Dec. 2009-Feb. 2010 (left panels) and for Dec. 1982-Feb. 1983 (right panels), denoting the southern end of the NAO. The analysis is done using the CPC standardized NAO (Jan. 1950 to present) and the Era-Interim reanalysis (Jan. 1979 to present)



subtropical jet. Superimposing on the impact of the El Niño Modoki, the downstream extension of the negative NAO largely strengthened and maintained the East Asian atmospheric circulation anomaly
responsible for the 2009/2010 drought, denoted by a much smaller RMS difference of the NAO points from the REF points than from the PEMI points (see the Fig. 7).

Differing from the 2009/2010 case, the 1982/1983 canonical El Niño exerted a dominant effect to shift the entrance of the Asian subtropical jet southward far from the Mediterranean Sea-the area of the optimal vorticity source for Rossby wave train (see the right panels of Fig. 11). Besides, the southern
end of the 1982/1983 NAO shifted westward from its normal position, without an eastward extension as far as the Mediterranean Sea. These features suggested the adverse conditions for the downstream extension of the NAO. Matching with a statistically insignificant correlation with the PNINO3 index (-0.11, at 95% significant confidence level ($p < 0.05$)), the occurrence of the positive NAO tended to be irrelevant to the occurrence of the canonical El Niño. The irrelevance, to some degree, was hinted at by
the orthogonal characteristic in the phase and structure of between the anomalous atmospheric circulation over the Middle East and East Asia related to the NAO and the canonical El Niño (see Fig. 6). In this regard, the downward extension of the NAO tended to be further obscured by the superimposition of the canonical El Niño-induced remote Rossby wave response in the atmosphere over the East Asia. As a result, a closer resemblance of the canonical El Niño-related pattern to the observed
pattern confirmed the leading role of the canonical El Niño in the relevant anomalous atmospheric circulation over the East Asia for the 1982/1983 wet case.

## 5. Summary and discussion

### 5.1 Summary

In the present research, we studied the potential physical processes to explain the formation of the
large-scale anomalous atmospheric circulation responsible for the 2009/2010 drought and 1982/1983 wet cases in the southern part of China, which aroused widespread public concern for the devastating societal and economic losses. As the highly anomalous forcing to the East Asian atmospheric circulation, a better understanding of how the ENSO and the NAO, both having a significant magnitude during the periods of the two cases, worked together to establish the moisture anomaly over the southern part of
China was focused on in the study.

The 2009/2010 drought was attributed to the combined impact of the El Niño Modoki and the negative NAO. Induced by the remote and local Rossby wave responses in the atmosphere to the El Niño Modoki-related condensational heat sink over the SCS, a weak anomalous ridge of high pressure in the upper troposphere appeared over the Indochina Peninsula, and a more zonal stretched anomalous
anticyclone in the lower troposphere was located over the SCS to the eastern Indian Ocean. In addition,



the southern end of the negative NAO steadily predominated over the North Atlantic Ocean as far as the Mediterranean Sea and exerted a dominant effect to shift the entrance of the Asian subtropical jet northward nearer to the Mediterranean Sea, and, by the Asian subtropical jet, effectively caused a stationary Rossby wave train including an upper troposphere anomalous ridge of high pressure

predominating over the Indochina Peninsula and an anomalous cyclone over the North and Northeast China in the lower and upper troposphere. So far, superimposing on the anomalous atmospheric circulation related to the El Niño Modoki, the downstream extension of the negative NAO largely strengthened and maintained the relevant anomalous atmospheric circulation for the 2009/2010 drought period. Under the circumstances, the dry conditions in the Southwest China occurred coinciding with

the weakened water vapor flux from the BOB, the divergence of water vapor and the anomalous northerly wind carrying drier air.

On the contrary, the 1982/1983 canonical El Niño exerted a dominant effect to shift the entrance of the Asian subtropical jet southward far from the Mediterranean Sea, and the southern end of the NAO moved westward far from the Mediterranean Sea‒ the area of the optimal vorticity source for Rossby

wave train. Under the circumstances, a continuous wave activity flux to the East Asia was not found, and the physical connection between the NAO and the East Asian atmospheric circulation anomaly was not steadily established. In addition, being irrelevant to the canonical El Niño, the downward extension of the NAO tended to be further obscured by the impact of the canonical El Niño over the East Asia. As a result, induced by the remote and local Rossby wave responses in the atmosphere to the canonical El

Niño-related condensational heat sink over the western Pacific Ocean, an upper troposphere anomalous cyclone was set up over the East Asia, and adjacent to the south was the anomalous Philippine Sea anticyclone. In this regard, the wet conditions in the South China occurred coinciding with a strengthened water vapor flux form the BOB and SCS and the induced convergence of water vapor at the bottom of the upper troposphere anomalous cyclone (i.e. at the upper troposphere trough). Therefore,

the strong canonical El Niño played a leading role in the relevant atmospheric circulation anomaly over the East Asia during the 1982/1983 case period.

The devastating drought or wet conditions in a region occurred due to the continuous dry or wet weather for a relatively long term. As pointed out in Trenberth and Guillemot (1996), modeling studies confirm the important role of antecedent soil moisture conditions and local feedbacks in perpetuating and

prolonging the moisture anomalies throughout the drought or wet periods. Therefore, a similar physical mechanism should be expected as the formation of the really dry land surface conditions for the 2009/2010 drought and the really wet ground surface situations for the 1982/1983 wet cases. The sensitivity experiment with numerical modeling simulation should be carried out in further studies on the impact of the antecedent soil moisture anomalies on the two cases.



## 5.2   Discussion

Fig.12 showed the general relationship for the precipitation indices in China, the NAO, PEMI, and

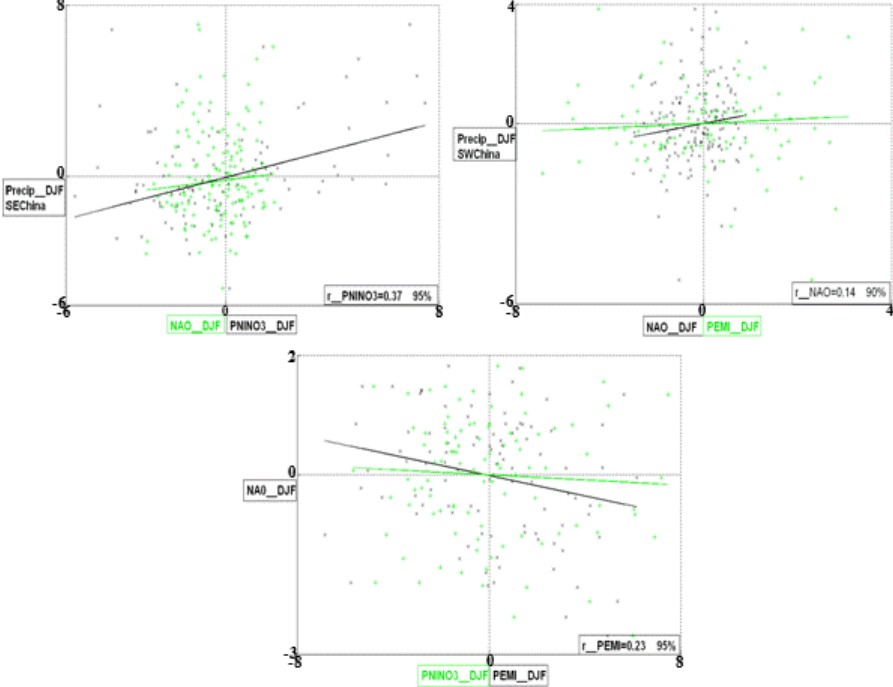

**Fig. 12** The scatter graph in the DJF for the precipitation index (Precip_DJF SEChina) of the Southeast China with the NAO and PNINO3 indices (upper left panel) and for the precipitation index (Precip_DJF SWChina) of the Southwest China with the NAO and PEMI indices (upper right panel), as well as for the NAO index with the PNINO3 and PEMI indices (lower panel). The PEMI and PNINO3 indices have a scale factor of 1e+6 and the precipitation index a scale factor of 10 for a similar scale of the x- and y-axis. During DJF, the precipitation index of the Southwest China is represented as the third principal components in China precipitation anomaly; and the precipitation index of the Southeast China is represented as the first principal components in China precipitation anomaly; both (accounting for 47% (the first principal components) and 3% (the third) of the total variance, respectively) are statistically significant in terms of the sampling error bars according to the rule proposed by North et al. (1982). In each panel, the two indices with the statistically significant correlation are represented by the scatter points in the black color and the green ones for the insignificant correlation, and in the lower right corner of each panel, the significant correlation coefficient between the two indices and the corresponding confidence interval are denoted. The solid black and green lines in each panel denote the linear regression curves between the two indices. The anomaly is defined as the deviation from the 30-year climatology (1980-2009). The analysis is done using the CPC standardized NAO (Jan. 1950 to present), the Era-Interim reanalysis (Jan. 1979 to present) and the Chen's precipitation in China (Jan. 1961 to Dec. 2010)





PNINO3 indices during boreal winter months (DJF). For the Southeast China (upper left panel) (as the 1982/1983 precipitation anomaly pattern), the canonical El Niño played a leading role in the precipitation anomaly (positive correlation), and the NAO had a minor impact in this region. However, for the Southwest China (upper right panel) (as the 2009/2010 precipitation anomaly pattern), the NAO played a major role in the precipitation anomaly (positive correlation), and the El Niño Modoki seemed not to make an important influence in the region. Actually, the lower panel of Fig. 12 further showed that the substantial impact of the El Niño Modoki in the Southwest China could be made by the significant relationship with the NAO, that is, the warm El Niño Modoki is associated with the negative NAO, and the negative NAO associated with the precipitation shortage in the Southwest China—the 2009/2010 drought pattern. The lower panel also showed that the NAO had an insignificant relationship with the canonical El Niño, which indicated the independent feature between them and further confirmed the insignificant relationship between the NAO and the precipitation anomaly in the Southeast China (upper left panel). Note that the asymmetric relationship of the NAO with the two types of the ENSO and their respective influences on the China precipitation anomaly were well characterized in the 2009/2010 and 1982/1983 cases.

On the basis of the present work, the downstream extension of the negative NAO tended to strengthen coinciding with the El Niño Modoki, but the canonical El Niño tended to weaken the downward extension of the positive NAO. Inferentially, the specific type of the ENSO determined if the NAO in the specific phase could exert a strong effect on the atmospheric circulation over the East Asia and its vicinity. On time scales of a few months and a few years, the ENSO makes an extraordinary impact on the global weather and climate. Recognized as one of the major features of the global climate system over the Northern Hemisphere, the NAO exerts a strong influence on the Northern Hemispheric weather and climate especially during the boreal winter. Therefore, the educated guess was an interesting issue and was worth a further study for the East Asian weather and climate anomalies.

Besides the ENSO and NAO, the persistent precipitation anomalies over the East Asia might be attributed to other potential factors, such as intraseasonal oscillations, storm tracks, the Tibetan Plateau, antecedent soil moisture conditions, local feedback and land processes, probably including the Indian Ocean SSTA. Due to a specific combination of these factors, each catastrophic wet or drought case had a unique cause. To understand better the particular physical processes responsible for each high impact weather and climate event, it was important to carry out the case study, which could improve the prediction skill of natural hazards to a certain extent.

**Acknowledgements**. This work was supported by the Chinese National Science Foundation Grant 40605019, and the National Program on Key Basic Research Project (973 Program) (2012CB417200). The ERA-Interim Reanalysis was provided from the ECMWF data server at http://data-protal.ecmwf.int.



The NCEP/NCAR Reanalysis derived data was supplied by the NOAA/OAR/ESRL PSD, Boulder, Colorado, USA, from their web site at http://www.esrl.noaa.gov/psd/. The NOAA_ERSST_V3 data was distributed by the NOAA/OAR/ESRL PSD, Boulder, Colorado, USA, from their web site at
http://www.esrl.noaa.gov/psd/. The National Meteorological Information Center of China provided the 752-station observations of 24-h accumulated precipitation in China at http://new-cdc.cma.gov.cn:8081/home.do/. In addition, Chen's latest grid precipitation in China was obtained from the website at http://rcg.gvc.gu.se/data/ChinaPrecip/DataandPlot.htm, and the CPC index of the NAO was gotten from the website at http://www.cpc.ncep.noaa.gov/data/teledoc/nao.shtml.

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
