# Peer review of "Role of NAO and ENSO in the anomalous precipitation in the southern part of"

_Natural Hazards and Earth System Sciences, 2017_

## Referee Comment (RC1) · Anonymous Referee #1 · 15 Nov 2017

Review of the research article (nhess-2017-143): "Role of NAO and ENSO in the anomalous precipitation in the southern part of China – study on the two contrary high impact weather and climate cases" by Qiuxia Wu.

This study highlights the socio-economic importance of anomalous dry and wet episodes concurrent with El Nino-Southern Oscillation (ENSO) and North Atlantic Oscillation (NAO) signatures in the background. The authors have presented the case with two significant episodes occurred during 1982/1983 (coincident with canonical ENSO and positive phase of NAO) and 2009/2010 (coincident with El Nino-Modoki and negative phase of NAO), and demonstrated how specific phase of NAO works with

ENSO pattern to produce anomalous dry and wet conditions over the study region. I must admit that this is a commendable effort and the results emerge from this study are interesting and very useful for improving prediction skill as well as for the long-term climate analysis. I certainly would recommend this article for publication subjected to a few minor concerns:

(1) The events considered in this study are hand-picked ones, how robust the conclusions would fit for similar case situations as addressed in this study. Are there any other years or events with combination of ENSO/NAO situations occurred over China before such as the ones used in this study. I don't see any mention about this in the text. How can we generalize the outcomes of this study? I understand it is going to be a big task, although a few sentences are mentioned (Lines 430-435), this rationale needs to be further be highlighted for the readers about the above.

(2) Given the gravity of this study and results, the dynamical scenario associated with the two events keeps repeating in many places, it needs to be made concise and precise. The authors need to work on reducing the redundancy. The text content is amply long and this primarily needs to be looked at, the manuscript needs substantial editing of texts and shortening of text so that the essence of the results is more visible to the readers.

(3) The contents of the figures (e.g., Figure 3; Lines 83-100) are generally not very clear (means to all figures), and I was not able to assess the contents in tune with the text. Labelling (a,b,c,d) for the sub-figures will be easier and the corresponding can be referred in the text to ease the reading. Also the figure captions are too long.

(4) Although overall grammar is good, the narration appears more of conversing nature at certain places, though it is a minor concern which can be revised easily. The author is seriously advised to sit with experts for grammar (right from beginning to the end of the manuscript for the flow to be in one tense) – I see that it keeps changing a lot. Some minor things I quote below, although the entire text needs a thorough check.

a. Line 79: "Thus, we hypothesize that the NAO works together….." b. Line 83: should it be "in accordance with…" c. Line 91: "However, a thorough investigation has not been carried out on how the two factors worked out to set up the relevant large-scale atmospheric circulation anomaly which is the focal point of this study" d. Line 135: "…calculated following Takaya et al.(2001) as follows: " (no need to mention C5 here). e. Line 140: "represents" in place of represented. f. Line 167: "influencing" in place of "affected" g. Line 246: Remove "to" h. In most places, the word "tended" can be "tend" or "tends" (e.g., Line 290) i. Concerning the lines referring the subtropical jet, you refer this as "entrance region of the Asian subtropical jet" … not as "entrance of the subtropical jet" appears to be vague. j. Line 425: "…an educated guess"

---

## Referee Comment (RC2) · Anonymous Referee #2 · 17 Nov 2017

The paper focused on the two cases of abnormal precipitation conditions over the southern China. The interactions of atmospheric circulations such as ENSO, NAO, Asian subtropical jet, Rossby waves and their roles in generating cyclones and anticyclones over the northern hemisphere were examined to identify the major reasons of the abnormal drought and wet conditions in the southwest and south China. Although these are useful knowledge, the represenativeness of these large scale circulation settings is however questionable and I don't know that only two cases would help to predict such abnormal conditions as the author claimed. I think that these case study probably should go to Monthly Weather Review or journals about weather which are perhaps more suitable. Another issue with the paper is English and the structure of the paper.

There are a lot minor issues including using abbreviation without full names, the variance of PC1 and PC4 are different in Introduction and Figure 12 caption; some figure captions are incomplete, too many figures etc.

---

## Author Comment (AC1) · 21 Nov 2017

Dear Referr 1:

It's pleasure for us to have a chance to get your comments for this manuscript!

Question 1: The events considered in this study are hand-picked ones, how robust the conclusions would fit for similar case situations as addressed in this study. Are there any other years or events with combination of ENSO/NAO situations occurred over China before such as the ones used in this study. I don't see any mention about this in the text. How can we generalize the outcomes of this study? I understand it is

going to be a big task, although a few sentences are mentioned (Lines 430-435), this rationale needs to be further be highlighted for the readers about the above.

Just as mentioned in line 430-435: "Due to a specific combination of these factors, each catastrophic wet or drought case had a unique cause. To understand better the particular physical processes responsible for each high impact weather and climate event, it was important to carry out the case study, which could improve the prediction skill of natural hazards to a certain extent.", that is, because each high impact case had a unique cause, this paper is focused on the case study on the two specific cases, not other cases, also the title of this paper made a clear description of the target of this paper. So, how can we generalized the outcomes of this study is beyond the task of this paper. Moreover, the two specific events in this study are not hand-picked ones, the line 30-35 made a clear introduction about their economic and societal importance for their high impact quality.

Question 2: Given the gravity of this study and results, the dynamical scenario associated with the two events keeps repeating in many places, it needs to be made concise and precise. The authors need to work on reducing the redundancy. The text content is amply long and this primarily needs to be looked at, the manuscript needs substantial editing of texts and shortening of text so that the essence of the results is more visible to the readers.

The necessary repeating of the dynamical scenario associated events is highly permitted and favorable to make a clear explanation for the dynamical scenario of the relative physical processes, it is suitable and not redundancy.

Question 3: The contents of the figures (e.g., Figure 3; Lines 83-100) are generally not very clear (means to all figures), and I was not able to assess the contents in tune with the text. Labelling (a,b,c,d) for the sub-figures will be easier and the corresponding can be referred in the text to ease the reading. Also the figure captions are too long.

The Figure 3 caption had a necessary and suitable description for this figure, it makes easier for readers to understand what the figure is saying, so it is not too long. Why not to carefully read the caption of Figure 3, then that would provide you a better understanding of line 83-100.

Question 4: Although overall grammar is good, the narration appears more of conversing nature at certain places, though it is a minor concern which can be revised easily. The author is seriously advised to sit with experts for grammar (right from beginning to the end of the manuscript for the flow to be in one tense) – I see that it keeps changing a lot. Some minor things I quote below, although the entire text needs a thorough check. a. Line 79: "Thus, we hypothesize that the NAO works together....." b. Line 83: should it be "in accordance with..." c. Line 91: "However, a thorough investigation has not been carried out on how the two factors worked out to set up the relevant large-scale atmospheric circulation anomaly which is the focal point of this study" d. Line 135: "...calculated following Takaya et al.(2001) as follows: " (no need to mention C5 here). e. Line 140: "represents" in place of represented. f. Line 167: "influencing" in place of "affected" g. Line 246: Remove "to" h. In most places, the word "tended" can be "tend" or "tends" (e.g., Line 290) i. Concerning the lines referring the subtropical jet, you refer this as "entrance region of the Asian subtropical jet" ... not as "entrance of the subtropical jet" appears to be vague. j. Line 425: "...an educated guess"

About a. Line 79:"Thus, we hypothesize that the NAO works together . . . . . .", I checked the manuscript and found that the line 79 in my manuscript is "Thus, we assumed that the NAO worked together . . . . . .", so I think you mean the word "hypothesize" is better than "assumed"? Actually, I checked the dictionary, saying that the two words means "to think that something is true but without having proof of it", that is, they have a similar meaning. Thus, this one is not better than that one. Moreover, in scientific paper, the past tense should be used to give a description of our work. So, the word "assumed" is correct.

About b. Line 83: should it be "in accordance with . . .", I checked the manuscript

and found that line 83 in my manuscript is "pressure in the upper troposphere and an induced strong subsidence of air according with a convergence", you mean "in accordance with" is better than "according with"? I checked the dictionary, saying that "in accordance with" means "according to a rule or the way that sb says that sth should be done", and "according with" means "to agree with or match sth". So I think my word is better.

About c. Line 91: "However, a thorough investigation has not been carried out on how the two factors worked out to set up the relevant large-scale atmospheric circulation anomaly which is the focal point of this study", I checked the manuscript and found that line 91 in manuscript is "However, no anyone did a thorough study on how the two factors worked together to set up the relevant large-scale atmospheric circulation anomaly, which was focused on in the study." You mean your sentence structure is better than mine? I don't think so!

About d. Line 135: "...calculated following Takaya et al.(2001) as follows: " (no need to mention C5 here), I checked the manuscript and found that line 135 in my manuscript is "calculated using the formulation of Eq. (C5) in Takaya et al. (2001) as follows:". First, Takaya et al. (2001) mentioned more than one equation to calculate of the wave activity flux, so it is necessary to mention which specific equation was used by us, and our expression is OK.

About e. Line 140: "represents" in place of represented.", I want to say that, in scientific paper, the past tense should be used when we talked about our work. So, "represented" is OK.

About f. Line 167: "influencing" in place of "affected", I checked the dictionary, saying that "affect" means "make a change in sth", and often passive, so, "affected" is OK. About g. Line 246: Remove "to", I want to say that the word "to" should be necessary in here!

About h. In most places, the word "tended" can be "tend" or "tends" (e.g., Line 290),

the answer same to e, that is, the past tense should be used in the description of our work. So, "tended" is OK.

About i. Concerning the lines referring the subtropical jet, you refer this as "entrance region of the Asian subtropical jet" ... not as "entrance of the subtropical jet" appears to be vague. I checked the manuscript from the beginning to the end, and I found the phrase "subtropical jet" appears 13 times, and every time they are appearing as a structure like "the Asian subtropical jet". So there is no vagueness in this paper.

About j. Line 425: "...an educated guess", I don't know what you mean? I checked the dictionary, saying "a guess that is based on some degree of knowledge, and is therefore likely to be correct".

Thanks for your hard work on this manuscript!

Best Wishes

Qiuxia Wu

---

## Author Comment (AC2) · 21 Nov 2017

Dear Refree 2:

It's pleasure for us to have a chance to get your comments for this manuscript again!

Question 1: Although these are useful knowledge, the represenativeness of these large scale circulation setting is however questionable and I don't know that only two cases would help to predict such abnormal conditions as the author claimed. I think that these case study probably should go to Monthly Weather Review of journals about weather which are perhaps more suitable.

[Figure]

First, the two cases are highlighted under the climate background, and the corresponding abnormal physical fields are statistically significant referring to the 30-year climatology of 1980-2009. Moreover, climate could be seen as the accumulation of weather for a time span. So, I think that climate and weather should not be seen as two completely independent concepts, they are closely linked to each other to some extent. Therefore, this manuscript might be suitable to MWR or Weather, but it indeed is suitable to NHESSD.

Second, this manuscript's title is "Role of NAO and ENSO in the anomalous precipitation in the southern part of China – study on the two contrary high impact weather and climate cases", so we just tried to explain the role of NAO and ENSO in the climate anomalies for the two specific cases. We did not make any prediction of such abnormal conditions in other cases. Could you please read the paper of Trenberth and Guillemot (1996) with the title "Physical processes involved in the 1988 drought and 1993 floods in North America".

Question 2: Another issue with the paper is English and the structure of the paper. There are a lot minor issues including using abbreviation without full names, the variance of PC1 and PC4 are different in Introduction and Figure 12 caption; some iňĄgure captions are incomplete, too many iňĄgures etc.

About this paper's language and structure, if you think they are not perfect, please make a clear detailed list for this issue. For example, which line, sentence or word is not good, and so on, and why, and what you think is good one? and why? Also about the structure of this paper, why you think our structure is not good, and what you think is good one, and why?

About abbreviation without full names, I could be certain to say that each abbreviation has full names in this paper!

About the difference of PC1 and PC4 between Introduction and Figure 12 caption, I checked the two figures. Figure 12 caption had clearly made a definition of principal

components of precipitation field for the winter (DJF). Figure 2 caption told us the analyzed period is Jan. 1961 to Dec. 2010, that means all seasons, including winter. So the difference between them is caused by the different analyzed periods of data. About too many figures, I want to say that, no too many figures, this paper selected the suitable figures for demonstration. Could you give us a clear explanation about which figures you think should be omitted, and why, and how to deal with the corresponding information of this figure in the suitable part of this paper?

Thanks for your hard work on this manuscript!

Best Wishes

Qiuxia Wu